# Insights from the Endophytic Fungi in *Amphisphaeria* (Sordariomycetes): *A. orixae* sp. nov. from *Orixa japonica* and Its Secondary Metabolites

**DOI:** 10.3390/microorganisms11051268

**Published:** 2023-05-11

**Authors:** Xiaojie Wang, Dhanushka N. Wanasinghe, Jingyi Zhang, Jian Ma, Peifeng Zhou, Lijuan Zhang, Yongzhong Lu, Zhen Zhang

**Affiliations:** 1School of Liquor and Food Engineering, Guizhou University, Guiyang 550025, China; wangjialian49@163.com (X.W.); gs.pfzhou20@gzu.edu.cn (P.Z.); 2Guizhou Academy of Testing and Analysis, Guizhou Academy of Sciences, Guiyang 550014, China; 3Centre for Mountain Futures, Kunming Institute of Botany, Chinese Academy of Sciences, Honghe County 654400, China; dnadeeshan@gmail.com; 4School of Food and Pharmaceutical Engineering, Guizhou Institute of Technology, Guiyang 550003, China; zjingyi127@gmail.com (J.Z.); yanmajian@163.com (J.M.); ljzhang@git.edu.cn (L.Z.)

**Keywords:** *Amphisphaeriales*, fungal taxonomy, hyphomycetes, isocoumarin, phylogeny

## Abstract

Endophytic fungi are a remarkably diverse group of microorganisms that have imperceptible associations with their hosts for at least a part of their life cycle. The enormous biological diversity and the capability of producing bioactive secondary metabolites such as alkaloids, terpenoids, and polyketides have attracted the attention of different scientific communities, resulting in numerous investigations on these fungal endophytes. During our surveys of plant-root-based fungi in the mountain areas of Qingzhen, Guizhou Province, several isolates of endophytic fungi were identified. In this study, a novel endophytic fungus was discovered in the roots of a medicinal plant (*Orixa japonica*) in Southern China and introduced as a new species (*Amphisphaeria orixae*) based on morphological evidence and molecular phylogenetic analysis (combined ITS and LSU sequence data). To the best of our knowledge, *A. orixae* is the first reported endophyte as well as the first hyphomycetous asexual morph in *Amphisphaeria*. A new isocoumarin, (*R*)-4,6,8-trihydroxy-5-methylisochroman-1-one (**1**), and 12 known compounds (**2**–**13**) were isolated from the rice fermentation products of this fungus. Using 1D- and 2D-NMR, mass spectrometry, and ECD studies, their structures were identified. The antitumor activity of these compounds was tested. Unfortunately, none of the compounds tested showed significant antitumor activity.

## 1. Introduction

Endophytic fungi are usually found in the internal tissues or organs of plants, but they do not cause noticeable tissue damage or symptoms. These fungi are often found in a wide array of plant species. At times, endophytic fungi are able to convert to a saprotrophic lifestyle when the host plant undergoes senescence [1]. Even though endophytic fungi can be found in a broad range of hosts, some fungi may be specific to certain hosts and have been extensively researched to gain an understanding of their biological properties. These fungi have also been studied as a source of novel and natural bioactive compounds. Each plant contains one or more endophytic microorganisms that have the potential to produce compounds similar to those of the host plant, and the secondary metabolites they produce are often characterized by novel structures that make them a hot research topic in the field of natural products [2,3]. As new microbial resources, endophytic fungi are an important source of natural bioactive products. The exploitation of endophytic fungi can help alleviate the problems of plant resource shortages and ecological imbalances. Additionally, the use of endophytic fungi can aid in the conservation of rare and endangered plant resources, making this research significant [4]. For these reasons, it is important to explore the secondary metabolites produced by endophytic fungi and to understand their potential applications.

*Orixa japonica* is a plant of the family *Rutaceae* and mainly distributed in China, Korea, and Japan [5]. It is a traditional folk medicine that is frequently used to cure diseases or ease symptoms, such as clearing heat and dampness, relieving cough, and acting as an analgesic for stomach pain and rheumatic joint pain [6]. Its extract has pharmacological and biological activities [7]. Liu et al. [8] obtained a new pyrrolizidine alkaloid from the ethanol extract of the root bark of *O. japonica*. The compound exhibited larvicidal activity against the fourth-instar larvae of *Aedes aegypti*, *Anopheles sinensis*, and *Culex pipiens pallens*, and it also possessed nematocidal activity against *Bursaphelenchus xylophilus* and *Meloidogynein congnita*. Choe et al. [9] reported that the leaf extract of *O. japonica* showed significant antimicrobial activity against *Bacillus cereus*, *Pseudomonas aeruginosa*, *Staphylococcus aureus*, and *Streptococcus mutans*. Huang et al. [10] isolated eighteen quinoline alkaloids from the roots of *O. japonica*, and some of the compounds showed significant anti-pathogenic fungal activity, with one compound demonstrating superior efficacy to that of the positive control hymexazol. Moreover, *O. japonica* can be used in conjunction with antibiotics such as florfenicol or amoxicillin to eliminate persistent cells. This may be attributable to the synergistic action of one or more compounds within the plant [11]. Domestic and international research on *O. japonica* has focused on its chemical composition, and a variety of chemicals have been isolated and identified from the *O. japonica* plant in previous studies [8]. In contrast, studies on endophytic fungi of *O. japonica* are relatively scarce or neglected. From the stem of *O. japonica*, Lu et al. [6] isolated the novel endophytic fungus *Diaporthe orixae* and discovered a new host record for *Diaporthe caryae*. At the same time, Japanese scholars similarly obtained an endophytic fungus belonging to *Diaporthe* from the leaves of *O. japonica* and inoculated it on MEA medium to obtain six polyketide derivatives as secondary metabolites from large-scale fermentation [12]. Polyketides, alkaloids, anthraquinones, and other types of novel metabolic products have been found in this genus, and many of them have significant antitumor, antibacterial, antioxidant, and other biological activities [13].

Since 2020, our research group has been focusing on the diversity of endophytic fungi of *Orixa japonica* [6]. In this study, a new endophytic fungus, *Amphisphaeria orixae*, was isolated from *O. japonica* and is introduced based on multi-gene phylogenetic analyses and morphological evidence. *A. orixae* is the first endophytic fungus of *Amphisphaeria* that forms a hyphomycetous asexual morph and is unique in undergoing thallic or blastic conidiogenesis and having polymorphic conidia. Thirteen secondary metabolites were identified from the rice fermentation product of *A. orixae*, including a new isocoumarin, (*R*)-4,6,8-trihydroxy-5-methylisochroman-1-one (**1**), and 12 known compounds (**2–13**). Their chemical structures are depicted in Figure 4 and were established using 1D- and 2D-NMR, mass spectrometry, and chemical computations.

## 2. Materials and Methods

### 2.1. Plant Material

The fresh and healthy whole plant of *Orixa japonica* was collected from Jiulong Mountain, Qingzhen City, Guizhou Province, China (106°30′7″ E 26°40′36″ N). Jiulong Mountain is located in the middle of the Yunnan–Guizhou Plateau, with a hilly landscape and a subtropical monsoonal humid climate, with an average annual temperature of 20 °C. For ease of transport to the laboratory and labeling with relevant metadata (including the date, habitat, location, and host), the materials were enclosed in sealed bags. Fungal isolation was carried out within 24 h of collection.

### 2.2. Isolation of Fungal Endophytes

The fresh and healthy materials were washed with running tap water for at least 10 min. The materials were surface-sterilized to eliminate epiphytic microorganisms in a benchtop by immersing them in 75% (*v*/*v*) ethanol for 3 min, then rinsed with sterilized distilled water for 2 min, then soaked with 10% (*v*/*v*) NaClO for 2 min, and finally rinsed with sterile distilled water three times continuously. The materials were dried on sterilized filter papers and then cut into small cubes (ca. 3 mm long segments) and placed on fresh potato dextrose agar (PDA) containing an antibiotic (50 μg/mL penicillin). Samples were incubated in a constant-temperature incubator (28 °C). The plates were observed daily, and mycelia from the edges of fungal colonies were transferred to fresh PDA plates to obtain pure cultures. To induce spore production, polypropylene (PP) was mixed with PDA medium and inoculated with the fungal mycelium. The mycelium was gently swept with a sterilized brush, and variable-temperature incubation was applied alternatively between 24 h at 28 °C and 24 h at 4 °C [14]. Specimens of the dried cultures were deposited at the Herbarium of Guizhou Academy of Agricultural Sciences (Herb. GZAAS). Living cultures were deposited at the Guizhou Culture Collection (GZCC).

### 2.3. Morphological Study

Micro-morphological characters were photographed using an ECLIPSE Ni-U compound microscope (Nikon, Tokyo, Japan) fitted with an EOS 90D digital camera (Canon, Tokyo, Japan). Tarosoft (R) Image Frame Work was used to measure different morphological features (including conidiogenous cells, mycelia, conidia, and conidiophores), while Adobe Illustrator CS6 (Adobe Systems, San Jose, CA, USA) was used for the processing of figures and pictures.

### 2.4. DNA Extraction, PCR Amplification, and Sequencing

Using sterile scalpels, fresh mycelia of fungi were scraped. Genomic DNA was extracted from the scraped mycelium by using the Biospin Fungus Genomic DNA Extraction Kit (BioFlux, Shanghai, China) following the manufacturer’s instructions. Two gene regions were amplified with universal primers, viz., the internal transcribed spacer region of ribosomal DNA (ITS: ITS5/ITS4) [15] and the partial large subunit nuclear ribosomal DNA (LSU: LR0R/LR5) [16]. The method used for PCR amplification of ITS and LSU by polymerase chain reaction is described by Lu et al. [17]. The quality of PCR products was observed by using ethidium bromide staining in 1% agarose gel electrophoresis. Successfully amplified PCR products were sent to Sangon Biotech (Shanghai, China) for purification and sequencing (with the same primers).

### 2.5. Sequence Alignment and Phylogenetic Analyses

The raw forward and reverse reads were edited for ambiguous bases at both ends and assembled using DNASTAR Lasergene SeqMan Pro v.7.1.0 (44.1). In order to compare species and build databases, newly acquired sequences were utilized as queries in BLAST searches of the NCBI GenBank nucleotide database. Each dataset was aligned using MAFFT v.7 [18]. Trimal was used to trim each alignment [19]. Alignment was checked manually using BioEdit [20].

For phylogenetic inferences, two genetic markers, ITS and LSU, were applied (Table 1). The phylogenetic tree was extrapolated using 58 taxa according to recent publications [21,22,23,24]. Maximum likelihood trees (ML) were constructed using IQ-Tree v.2 [25]. The combined datasets were analyzed and divided into groups based on genetic markers. One thousand ultrafast bootstrap replicates were used to estimate branch support. MP analysis was performed with the CIPRES Science Gateway platform using the PAUP on XSEDE (4.a168) tool. ModelTest, implemented in MrMTgui [26], was used to identify the most suitable evolutionary model for Bayesian inference analysis using the Akaike Information Criterion (AIC). One thousand rapid bootstrap replicates were used to estimate bootstrap support. Posterior probabilities (PPs) were evaluated in MrBayes v.3.1.2 [27] with Markov chain Monte Carlo sampling (MCMC). The number of generations for each dataset was determined independently and is explicitly stated in the legend of each tree. The top 25% of the trees represent the aging stage of the analysis and were therefore discarded, while the rest of the trees were used to compute PPs for the majority-rule consensus tree. Convergence was declared for all Bayesian inference trees when the average standard deviation was 0.01. The FigTree v1.4.0 program was used to draw the figures of the trees [28]. The new family’s placement was evaluated using the approximately unbiased (AU) test implemented in CONSEL [29]. Topologies with *p* values less than 0.05 in the AU test were regarded as being rejected.

### 2.6. Fermentation, Extraction, and Separation

The strain was incubated on PDA medium at a constant temperature of 28 °C for 7 days, then cut into small pieces (about 3 × 3 mm) with a scalpel and transferred to a 250 mL conical flask containing 100 mL of liquid medium (20 g of maltose, 10 g of sodium glutamate, 0.5 g of KH_2_PO_4_, 20 g of mannitol, 0.3 g of MgSO_4_-7H_2_O, 3 g of yeast paste, 10 g of glucose, and 1 L of tap water), and incubated on a 28 °C shaker (150 rpm) for about 10 days. Then, 5 mL of the seed liquid that had been made with 50 g of rice and 55 mL of distilled water was transferred to a sterilized plastic container with a capacity of 200 mL. The sterilized bags were then cultured for 65 days at a constant 28 °C temperature. A total of 1000 bags were cultured.

The crude extract was obtained by extracting the fermented product in a 10:1 ratio of ethyl acetate to methanol and then removing the solvent under reduced pressure. The crude extract was first dissolved in a methanol solution, and then an equal amount of a petroleum ether solution was added, and the petroleum ether extract (212 g) and methanol extract (380 g) were obtained after partitioning. In order to enrich the alkaloids, the methanol layer of the extract was dissolved with an appropriate amount of 2% HCl solution, then the acidic water was extracted three times with an equal volume of ethyl acetate, the ethyl acetate layer was discarded, and the pH was adjusted to about 10 with 25% aqueous ammonia. The mixture was extracted three times with an equal volume of chloroform, and the chloroform layer was combined and concentrated to obtain 2 g of extract (Fr.1). The remaining solution was adjusted to neutral pH, concentrated under reduced pressure, and then separated by silica gel column chromatography (Qingdao Marine Chemistry Co. Ltd., Qingdao, China) with dichloromethane/methanol (100:0→0:100, *v/v*) to obtain six fractions (Fr.2–Fr.7). Fr.1 (2 g) was further separated using Sephadex LH-20 (Amersham Pharmacia Biotech AB) (MeOH) into three fractions (Fr.1.1–Fr.1.3). Fr.1.1 (500 mg) was further separated by semi-preparative HPLC (SHIMADZU Essentia LC-16P, GH0525010C18 column 10 × 250 mm, 5 µm) using CH_3_OH/H_2_O (55:45) to obtain compound **5** (4.5 mg, 3 mL/min, *t*_R_ = 18.1 min) and compound **13** (6 mg, 3 mL/min, *t*_R_ = 30.1 min), Fr.1.2 (320 mg) was further separated by semi-preparative HPLC using CH_3_OH/H_2_O (40:60) to obtain compound **10** (4.4 mg, 3 mL/min, *t*_R_ = 6.2 min) and compound **11** (3.2 mg, 3 mL/min, *t*_R_ = 7.19 min), and Fr.1.3 (70 mg) was further separated by semi-preparative HPLC using CH_3_CN/H_2_O (40:60) to obtain compound **12** (9.5 mg, 3 mL/min, *t*_R_ = 8.5 min). Fr.2 (20 g) was separated by silica gel column chromatography (200–300 mesh) to obtain three fractions (Fr.2.1–Fr.2.3). Fr.2.2 (4 g) was further separated by Sephadex LH-20 and semi-preparative HPLC using CH_3_OH/H_2_O (65:35) to obtain compound **6** (16 mg, 3 mL/min, *t*_R_ = 19.6 min), and Fr.2.3 (600 mg) was further separated by semi-preparative HPLC using CH_3_CN/H_2_O (38:62) to obtain compound **7** (150 mg, 3 mL/min, *t*_R_ = 8.6 min) and compound **8** (4.4 mg, 3 mL/min, *t*_R_ = 11.3 min). Fr.3 (20 g) was separated by repeated silica gel column chromatography (200–300 mesh) to obtain 5 fractions (Fr.3.1–Fr.3.5). Fr.3.2 (3 g) was separated by Sephadex LH-20 (MeOH) to obtain 4 fractions (Fr.3.2.1–Fr.3.2.4), and Fr.3.2.4 (510 mg) was further separated by semi-preparative HPLC using CH_3_OH/H_2_O (45:55) to obtain compound **2** (18 mg, 3 mL/min, *t*_R_ = 17 min), compound **3** (29 mg, 3 mL/min, *t*_R_ = 12.3 min), and compound **4** (18 mg, 3 mL/min, *t*_R_ = 8.5 min). Fr.5 (20 g) was separated by repeated silica gel column chromatography (200–300 mesh) to obtain 6 fractions (Fr.4.1–Fr.4.6), and Fr.4.2 (3 g) was further separated by Sephadex LH-20 (methanol elution) to obtain 4 fractions (Fr.4.2.1–Fr.4.2.4). Fr.4.2.4 (162 mg) was further separated by semi-preparative HPLC using CH_3_OH/H_2_O (55:45) to obtain compound **9** (11 mg, 3 mL/min, *t*_R_ = 19.2 min), Fr.4.4 (1.7 g) was further separated by Sephadex LH-20 (dichloromethane/methanol 1:1 ratio elution) to obtain 3 fractions (Fr.4.4.1–Fr.4.3), and Fr.4.4.2 (69 mg) was further separated by semi-preparative HPLC using CH_3_OH/H_2_O (30:70) to obtain compound **1** (36.6 mg, 3 mL/min, *t*_R_ = 11.6 min).

### 2.7. Antitumor Bioassay

The following established in vitro human cancer cell lines were used: human breast cancer cells (MCF-7), human hepatocellular carcinoma cells (HepG2), and pancreatic cancer cells (PANC-1). Adriamycin was used as a positive control, and antitumor activity was measured for compounds **1**, **3**, **4**, **7**, **9**, **12**, and **13**. Cell counting kit-8 was obtained from GLPBIO (USA). MCF-7 and PANC-1 cells in the logarithmic growth phase were digested and added to 96-well plates at a density of 0.6 × 10^4^ cells per well and incubated at 37 °C in a 5% CO_2_ incubator for 12 h. After the cells were plastered, culture solutions containing 0, 50, 100, and 200 µM of the drug at different concentrations were prepared by performing DMSO gradient dilution in a volume of 100 µL. Each group of five replicate wells was incubated in the cell culture incubator for 24 h; the plate was washed twice with 100 µL of PBS, and then 100 µL of culture solution containing 10% CCK-8 was added. The OD value was measured at a 450 nm wavelength. It was calculated by the following equation.
Cell survival rate (%) = [(As − Ab)/(Ac − Ab)] × 100.

As = absorbance of experimental wells (containing cells, medium, CCK-8, and compound to be measured).

Ab = absorbance of blank wells (containing medium, and CCK-8).

Ac = absorbance of control wells (with cells, medium, and CCK-8).

## 3. Results

### 3.1. Phylogenetic Analysis of ITS and LSU Sequences

The aligned sequence matrix comprised ITS (617 bp) and LSU (850 bp) sequence data, including 56 ingroup taxa and two outgroup taxa. The dataset has 1467 characters, of which 884 characters are constant, 160 variable characters are quasi-informative, and 423 characters are quasi-informative. ML, MP, and Bayesian analyses of the combined dataset resulted in the reconstruction of plant genetics with essentially similar topologies, and the ML tree is shown in Figure 1. The tree is rooted with *Achaetominum macrosporum* (CBS 532.94) and *Chaetomium elatum* (CBS 374.66). The genus *Amphisphaeria* contained 25 taxa retrieved from GenBank and two new strains generated in this study. The two new isolates are recognized as one new species, viz., *Amphisphaeria orixae*, and it formed a sister clade with *A. camelliae* (HKAS 107021 and MFLUCC 20-0122) and *A. uniseptata* (CBS 114967).

### 3.2. Taxonomy

*Amphisphaeria* Ces. & De Not., Comm. Soc. crittog. Ital. 1(fasc. 4): 223 (1863).

Index Fungorum number: IF 173; Facesoffungi number: FoF 02099.

Type species: *Amphisphaeria umbrina* (Fr.) De Not, Sfer. Ital: 69 (1863).

Notes: *Amphisphaeria* is the type genus of *Amphisphaeriaceae*, with *A. umbrina* as the type species [22,23]. Most of the species of *Amphisphaeria* are found on grasses, woody branches, and some monocotyledons as saprobes in terrestrial habitats [30]. The unicellular ascospores of *Amphisphaeria* members typically have J+ or J− apical rings, solitary or aggregated ventral membranes under undeveloped perithecia or absent perithecia; and a coelomycetous asexual morphology with a light- to dark-brown color, oval to fusiform shape, and 1-3-noded ventral conidia [21,22,23,24]. The species of *Amphisphaeria* that have been reported are sexual morphs or coelomycetous asexual morphs, and no studies have reported hyphomycetous asexual morphs in this genus so far. In addition, Amphisphaeriaceous taxa have not been recorded as phytopathogens [30]. Based on phylogenetic analysis and morphological evidence, we introduce the first endophytic fungus of *Amphisphaeria* in this study. It is distinguished as a hyphomycetous asexual morph.

*Amphisphaeria orixae* X. J. Wang, Y. Z. Lu & Z. Zhang, sp. nov., Figure 2.

Index Fungorum number: IF 900170; Facesoffungi number: FoF 13911.

Etymology—Name referring to the host “*Orixa japonica*”, using the genitive case meaning “of Orixa”.

Holotype: GZAAS 22-2031.

Description: Endophytic in the roots of *O. japonica*. Sexual morph: Undetermined. Asexual morph: Two modes of development during conidiogenesis. (1) Thallic (d–n). Conidiophores hypha-like or reduced. Conidiogenous cells holothallic, narrowly cylindrical, frequently undifferentiated, and hyaline, forming conidia by random thallic-arthric disarticulation or alternate-arthric conidiogenesis. Conidia produced by thallic-arthric conidiogenesis, aseptate, micro guttulate, hyaline, polymorphic, bacilliform, cylindrical or cuneiform (2–13 × 1.5–2 µm, *n* = 23); conidia produced by alternate-arthric conidiogenesis, aseptate, guttulate, globose to subglobose (5.5–7.5 × 4.5–6.5 µm ($\overset{-}{x}$ = 6 × 5 µm, *n* = 12)], ellipsoidal 3–6.5 × 3–4.5 ($\overset{-}{x}$ = 4.5 × 3.5 µm, *n* = 18). (2) Phialidic (o–t). Conidiophores hyaline, penicillate, monoverticillate to biverticillate, with a stipe and apical series of branches subtending the phialides. Phialides 5.5–10 × 2–3 µm ($\overset{-}{x}$ = 7.5 × 2.5 µm, *n* = 15), lageniform to cylindrical, straight or curved, and hyaline. Conidia 2.8–3.5 µm diam, globose, aseptate, hyaline, and smooth when young, and pale yellow, verrucose, or echinulate when mature.

Material examined: CHINA, Guizhou Province, Qingzhen city, from the healthy roots of *O. japonica*, 14 September 2020, Hong-Bo Wang, Jg1 (Holotype: GZAAS 22-2031, dried culture), ex-type culture: GZCC 22-2031; *ibid*., Jg2 (GZAAS 22-2032); living culture: GZCC 22-2032.

Culture characteristics: Colonies whole, superficial, flat, slow growing on PDA, reaching 10 mm diam after 10 d at 28 °C, light yellow to brownish yellow with a clear brown irregular boundary in the middle. Vegetative hyphae septate, branched, hyaline, or yellow.

Notes: *Amphisphaeria orixae* is the first hyphomycetous asexual morph reported in this genus based on molecular DNA data, and it morphologically differs from all existing *Amphisphaeria* species. *Amphisphaeria camelliae* (HKAS 107021, MT756615) was the closest species to our newly obtained isolates based on the BLAST results of LSU (99.08% similarity). The phylogenetic analyses showed support values with 98% MLBS/1.00 PP (Figure 1) indicating that the two new isolates of *A. orixae* form a distinct clade within the genus and are a sister clade to *A. camelliae* (HKAS 107021 and MFLUCC 20-0122) and *A. uniseptata* (CBS 114967). Moreover, *A. orixae* is also the first endophyte in the genus *Amphisphaeria*.

### 3.3. Structure Elucidation

Compound **1** was isolated as a white powder, and its molecular formula was determined as C_10_H_10_O_5_ by high-resolution mass spectrometry (HRESIMS) (*m/z* 209.0465[M-H]^+^, calcd. for C_10_H_9_O_5_, 209.0455), indicating 6 degrees of unsaturation. UV spectra indicated the presence of a benzophenone moiety in the compound because of the maximum absorption at 267 and 314 nm. The IR spectrum indicated the existence of groups such as hydroxyl (3366 cm^−1^), carbonyl (1669 cm^−1^), benzene ring (1621, 1499 cm^−1^), and methyl (1468 cm^−1^). The ^1^H-NMR spectral data (Table 2) showed the presence of a single aromatic proton signal *δ*_H_ 6.35 (1H, s), a hydroxylated methine signal *δ*_H_ 4.89 (1H, dd, *J* = 2.4, 2.4 Hz), two oxygenated methylene signals (*δ*_H_ 4.59 (1H, dd, *J* = 12.6, 2.4 Hz) and 4.48 (1H, dd, *J* = 12.6, 2.4 Hz)), and a methyl signal *δ*_H_ 2.16 (3H, s). The ^13^C-NMR spectral data (Table 2) give a total of 10 carbon signals, consisting of one ester carbonyl, six aromatic carbons corresponding to a benzene ring, one oxygenated methylene carbon, one hydroxylated sp^3^ methine, and a methyl. These data were quite similar to those of the known analog (*R*)-4-acetyl-6,8-dihydroxy-5-methylisochroman-1-one (**4**), with the main difference being that signals for the sp^3^ methine C-10 and the attached acetyl substituent in **4** were replaced by signals for one hydroxylated sp^3^ methine at *δ*c_/H_ 62.1/4.89 in **1**, which indicates that a hydroxyl rather than an acetyl substituent as in **4** is located at C-10 of **1**. This deduction was further confirmed by COSY correlations of H-10/H_2_-9 and HMBC correlations (Figure 3A) from H-10 to C-4; H_2_-9 to C-5 and C-7; H_3_-11 to C-1, C-5, and C-6; and H-2 to C-1, C-3, and C-4. To ascertain the absolute configuration of **1**, The conformations of the isomers of compound **1** were generated using the iMTD-GC method embedded in the Crest program [31]. ECD calculations for **1** (10*S*) and *ent*-**1** (10*R*) were performed using Gaussian software, the results of which were compared with experimental values (detailed ECD calculations are in the Appendix A). As shown in Figure 3B, the experiment ECD curve matched well with the calculated curve for **1**, thus assigning its absolute configuration as 10*S*.

Compounds **2–13** were identified as known compounds based on their spectral features and by comparison with the literature (Figure 4): (3*R*,4*R*)-4-acetyl-3-ethoxy-6,8-dihydroxy-5-methylisochroman-1-one (**2**) [32]; (3*R*,4*R*)-4-acetyl-6,8-dihydroxy-3-methoxy-5-methylisochroman-1-one (**3**) [33]; (*R*)-4-acetyl-6,8-dihydroxy-5-methylisochroman-1-one (**4**) [34]; kokusaginine (**5**) [35]; dictamnine (**6**) [36]; lunidonine (**7**) [37]; orixinone (**8**) [38]; N-methylpreskimmianine (**9**) [39]; isoplatydesmine (**10**) [40]; *S*-ribalinine (**11**) [41]; (-)-(*S*)-edulinine (**12**) [42,43]; and orixalone A (**13**) [44], respectively. In the Appendix A, the NMR data for well-known compounds are described in detail.

**Figure 4 microorganisms-11-01268-f004:**
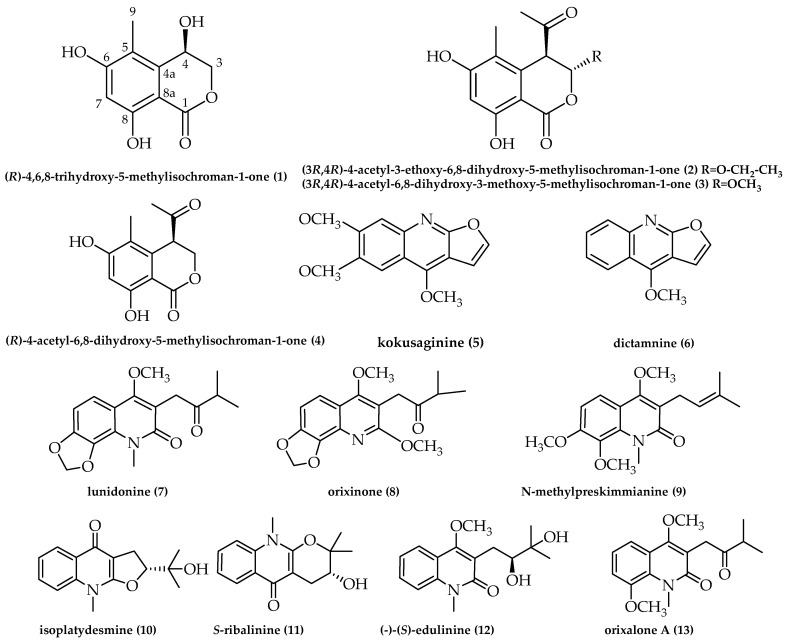
Chemical structures of compounds **1**–**13**.

Compound **2**, C_14_H_16_O_6_, yellow oil, HREIMS *m*/*z* 279.0879 [M-H]^+^ (calcd. for C_14_H_15_O_6_, 279.0862), ^1^H-NMR (600 MHz, MeOD, *δ*, ppm) *δ*_H_ 6.34 (1H, s, H-7), 5.83 (1H, d, *J* = 1.2 Hz, H-3), 4.30 (1H, d, *J* = 1.2 Hz, H-4), 3.91–3.88 (1H, m, H- 1’), 3.76–3.74 (1H, m, H-1’), 2.18 (3H, s, H-12), 2.04 (3H, s, H-13), 1.17 (3H, t, *J* = 7.2, 1.2 Hz, H-2’); ^13^C-NMR (150 MHz, MeOD, *δ*, ppm) *δ*_C_ 204.6 (C-11), 169.5 (C-1), 165.0 (C-5), 163.7 (C-8), 136.2 (C-10), 117.4 (C-5), 102.3 (-CH), 102.2 (-CH), 101.1 (C-9), 66.5 (-CH_2_), 55.0 (-CH), 29.1 (-CH_3_), 15.3 (-CH_3_), 11.1 (-CH_3_).

Compound **3**, C_13_H_14_O_6_, brown solid, HREIMS *m*/*z* 265.0722 [M-H]^+^ (calcd. for C_13_H_13_O_6_, 265.0722), ^1^H-NMR (600 MHz, DMSO-*d*_6_, *δ*, ppm), *δ*_H_ 6.36 (1H, s, H-7), 5.84 (1H, d, *J* = 5.4 Hz, H-3), 4.48 (1H, s, H-4), 3.45 (3H, s, H-11), 2.27 (3H, d, *J* = 1.8 Hz, H-13), 1.94 (3H, s, H-14); ^13^C-NMR (150 MHz, DMSO-*d*_6_, *δ*, ppm) *δ*_C_ 202.9 (C-12), 167.2 (C-1), 163.3 (C-6), 161.2 (C-8), 135.0 (C-10), 115.6 (C-5), 101.0 (C-3), 101.0 (C-7), 99.8 (C-9), 56.3 (C-11), 52.8 (C-4), 29.4 (C-13), 10.8 (C-14).

Compound **4**, C_12_H_12_O_5_, light-yellow crystal, ESI-MS *m*/*z* 259.1 [M + Na]^+^; ^1^H-NMR (600 MHz, DMSO-*d*_6_, *δ*, ppm), *δ*_H_ 11.00 (2H, s, H-3), 6.37 (1H, s, H-7), 4.93 (1H, dd, *J* = 11.4, 5.4 Hz, H-3a), 4.61 (1H, dd, *J* = 12, 4.2 Hz, H-3b), 4.28 (1H, d, *J* = 3 Hz, H-4), 2.28 (3H, s, H-12), 1.93 (3H, S, H-13); ^13^C-NMR (150 MHz, DMSO-*d*_6_, *δ*, ppm) *δ*_C_ 205.1 (C-11), 169.1 (C-1), 162.9 (C-6), 161.4 (C-8), 137.8 (C-10), 114.5 (C-5), 101.2 (C-7), 100.4 (C-9), 68.1 (C-3), 47.7 (C-4), 28.6 (C-12), 10.7 (C-13).

Compound **5**, C_14_H_13_NO_4_, white solid, ESI-MS *m*/*z* 260.1 [M + H]^+^; ^1^H-NMR (400 MHz, CDCl_3_, *δ*, ppm), *δ*_H_ 7.58 (1H, d, *J* = 2.8 Hz, H-2′), 7.48 (1H, s, H-5), 7.34 (1H, s, H-8), 7.05 (1H, d, *J* = 2.4 Hz, H-3′), 4.44 (3H, s, 4-OCH3), 4.03 (6H, d, *J* = 2 Hz, 6-OCH_3_, 7-OCH_3_); ^13^C-NMR (100 MHz, CDCl_3_, *δ*, ppm), *δ*_C_ 163.2 (C-2), 155.7 (C-4), 152.7 (C-6), 147.9 (C-7), 142.7 (C-8a), 142.6 (C-2′), 113.1 (C-3), 106.8 (C-8), 104.8 (C-3′), 102.3 (C-4a), 100.3 (C-5), 59.0 (4-OCH_3_), 56.2 (6-OCH_3_), 56.1 (7-OCH_3_).

Compound **6**, C_12_H_9_NO_2_, yellow crystal, ESI-MS *m*/*z* 200.1 [M + H]^+^; ^1^H-NMR (400 MHz, CDCl_3_, *δ*, ppm), *δ*_H_ 8.25 (1H, dd, *J* = 8.4, 0.8 Hz, H-5), 8.01 (1H, d, *J* = 8.4 Hz, H-8), 7.67 (1H, td, *J* = 6.8, 1.2 Hz, H-7), 7.60 (1H, d, *J* = 2.8 Hz, H-2), 7.43 (1H, td, *J* = 6.8, 1.2 Hz, H-6), 7.04 (1H, d, *J* = 2.8 Hz, H-3, 4.41 (3H, s, -OCH_3_); ^13^C-NMR (100 MHz, CDCl_3_, *δ*, ppm), *δ*_C_ 163.9 (C-9a), 156.9 (C-4), 145.7 (C-8a), 143.6 (C-2), 129.7 (C-7), 127.8 (C-8), 123.8 (C-6), 122.5 (C-5), 118.8 (C-3a), 104.8 (C-3), 103.4 (C-4a), 59.4 (-OCH_3_).

Compound **7**, C_17_H_19_NO_5_, white crystal, ESI-MS *m*/*z* 340.1 [M + Na]^+^; ^1^H-NMR (400 MHz, CDCl_3_, *δ*, ppm), *δ*_H_ 7.38 (1H, d, *J* = 8.8 Hz, H-5), 6.82 (1H, d, *J* = 8.4 Hz, H-6), 6.03 (2H, s, OCH_2_O), 3.85 (6H, d, *J* = 5.2 Hz, N-CH_3_, -OCH_3_), 3.78 (2H, s, H-11), 2.87 (1H, m, H-13), 1.22 (3H, s, CH_3_), 1.20 (3H, s, CH_3_); ^13^C-NMR (150 MHz, CDCl_3_, *δ*, ppm), *δ*_C_ 212.0 (C-12), 163.9 (C-2), 162.2 (C-4), 149.6 (C-7), 133.9 (C-8), 126.1 (C-10), 118.3 (C-5), 115.5 (C-3), 114.3 (C-9), 104.8 (C-6), 101.2 (OCH_2_O), 62.2 (OCH_3_), 41.3 (C-13), 37.0 (C-11), 32.8 (N-CH_3_), 18.5 (C-14, 15).

Compound **8**, C_17_H_19_NO_5_, colorless crystal, HREIMS: *m*/*z*: 340.1149 [M + Na]^+^ (calcd. for C_14_H_15_O_6_Na, 340.1149), ^1^H-NMR (600 MHz, MeOD, *δ*, ppm), *δ*_H_ 7.53 (1H, d, *J* = 8.4 Hz, H-4), 7.08 (1H, d, *J* = 8.4 Hz, H-3), 6.16 (2H, s, H-18), 3.98 (3H, s, H- 17), 3.92 (2H, s, H-11), 3.89 (3H, s, H-16), 2.87 (1H, m, H- 3), 1.19 (6H, d, *J* = 6.6 Hz, H-14, H-15); ^13^C-NMR (150 MHz, MeOD, *δ*, ppm), *δ*_C_ 214.4 (C-12), 164.9 (C-8), 163.9 (C-6), 149.1 (C-2), 141.7 (C-10), 134.0 (C-1), 118.8 (C-7), 117.2 (C-4), 110.9 (C-5), 108.5 (C-3), 103.3 (CH_3_), 63.1 (CH_3_), 54.4 (CH_3_), 41.9 (C-13), 37.0 (C-11), 18.8 (-CH_3_, -CH_3_).

Compound **9**, C_18_H_23_NO_4_, colorless crystal, ESI-MS *m*/*z* 318.3 [M + H]^+^; ^1^H-NMR (600 MHz, CDCl_3_, *δ*, ppm), *δ*_H_ 7.55 (1H, d, *J* = 9 Hz, H-5), 6.90 (1H, d, *J* = 9 Hz, H-6), 5.25 (1H, m, H-12), 3.95 (6H, d, *J* = 6 Hz, OCH_3_, OCH_3_), 3.87 (3H, s, OCH_3_), 3.77 (3H, s, N-CH_3_), 3.36 (2H, d, *J* = 6.6 Hz, H-11), 1.80 (3H, s, CH_3_), 1.68 (3H, d, *J* = 1.2 Hz, CH_3_); ^13^C-NMR (150 MHz, CDCl_3_, *δ*, ppm), *δ*_C_ 165.6 (C-2), 160.2 (C-4), 154.9 (C-7), 137.1 (C-8), 134.3 (C-13), 132.4 (C-10), 121.9 (C-12), 120.0 (C-3), 119.3 (C-5), 114.0 (C-9), 107.4 (C-6), 61.8 (OCH_3_), 61.8 (OCH_3_), 56.4 (OCH_3_), 34.1 (N-CH_3_), 25.9 (CH_3_), 24.4 (C-11), 18.1 (CH_3_).

Compound **10**, C_15_H_17_NO_3_, white crystal, ESI-MS *m*/*z* 282.1 [M + Na]^+^; ^1^H-NMR (600 MHz, MeOD, *δ*, ppm), *δ*_H_ 8.30 (1H, dd, *J* = 7.8, 1.2 Hz, H-5), 7.74 (1H, ddd, *J* = 1.8, 7.2, 8.4 Hz, H-7), 7.69 (1H, d, *J* = 8.4 Hz, H-8), 7.43 (1H, m, H-6), 4.98 (1H, td, *J* = 8.4, 2.4 Hz, H-3), 3.81 (3H, s, N-Me), 3.24 (2H, d, *J* = 8.4 Hz, H-2), 1.38 (3H, s, 2′-Me), 1.26 (3H, s, 3′-Me); ^13^C-NMR (150 MHz, MeOD, *δ*, ppm), *δ*_C_ 174.9 (C-4), 164.3 (C-3b), 140.0 (C-8a), 132.7 (C-7), 126.7 (C-4a), 126.4 (C-5), 124.8 (C-6), 116.5 (C-8), 101.7 (C-3a),93.5 (C-2), 72.2 (C-1′), 32.2 (N-Me),28.1 (C-3), 25.8 (C-3), 24.7 (C-2′).

Compound **11**, C_15_H_17_NO_3_, white crystal, ESI-MS *m*/*z* 282.1 [M + Na]^+^; ^1^H-NMR (600 MHz, MeOD, *δ*, ppm), *δ*_H_ 8.30 (1H, dd, *J* = 8.4, 1.2 Hz, H-6), 7.74 (1H, dd, *J* = 9, 1.8 Hz, H-7), 7.71 (1H, d, *J* = 8.4 Hz, H-9), 7.40 (1H, td, *J* = 6.6, 1.2 Hz, H-8), 3.88 (1H, t, *J* = 5.4 Hz, H-3), 3.79 (3H, s, N-Me), 2.95 (1H, dd, *J* = 16.8, 4.8 Hz, H-4b), 2.73 (1H, dd, *J* = 16.8, 6 Hz, H-4a), 1.48 (3H, s, 1′-Me), 1.46 (3H, s, 2′-Me); ^13^C-NMR (150 MHz, MeOD, *δ*, ppm), *δ*_C_ 178.4 (C-5), 156.9 (C-9b), 140.4 (C-9a), 133.2 (C-8), 126.5 (C-7), 124.5 (C-5b), 124.2 (C-6), 116.6 (C-9), 98.5 (C-5a), 84.3 (C-2), 69.0 (C-3), 31.2 (N-Me), 26.8 (C-4), 25.3 (C-1′), 22.1 (C-2′).

Compound **12**, C_16_H_21_NO_4_, white solid, HREIMS *m*/*z* 314.1359 [M + Na]^+^ (calcd. for C_16_H_21_NO_4_Na, 314.1359), ^1^H-NMR (600 MHz, MeOD, *δ*, ppm), *δ*_H_ 7.90 (1H, d, *J* = 7.8 Hz, H-5), 7.64 (1H, t, *J* = 8.4 Hz, H-7), 7.60 (1H, d, *J* = 9 Hz, H-8), 7.34 (1H, t, *J* = 7.2 Hz, H-6), 3.99 (3H, s, 4-OCH_3_), 3.74 (3H, d, *J* = 1.2 Hz, N-CH_3_), 3.70 (1H,dd, *J* = 10.2, 2.4 Hz, H-12), 3.03 (1H, dd, *J* =13.2, 1.2 Hz, H-11), 2.81 (1H, t, *J* = 10.2 Hz, H-10), 1.29 (3H, S, CH_3_), 1.28 (3H, s, CH_3_); ^13^C-NMR (150 MHz, MeOD, *δ*, ppm), *δ*_C_ 166.8 (C-2), 163.7 (C-4), 140.3 (C-10), 131.9 (C-7), 124.7 (C-5), 123.7 (C-6), 122.0 (C-9), 119.1 (C-3), 116.0 (C-8), 78.7 (C-12), 74.0 (C-13), 63.0 (4-OCH_3_), 30.5 (N-CH_3_), 28.6 (C-11), 26.0 (CH_3_), 25.0 (CH_3_).

Compound **13**, C_17_H_21_NO_4_, white powder, ESI-MS *m*/*z* 326.1 [M + Na]^+^; ^1^H-NMR (600 MHz, CDCl_3_, *δ*, ppm), *δ*_H_ 7.44 (1H, dd, *J* = 7.8, 1.2 Hz, H-5), 7.17 (1H, t, *J* = 7.8 Hz, H-6), 7.06 (1H, dd, *J* = 8.4, 1.2 Hz, H-7), 3.92 (3H, s, 1-NCH_3_), 3.88 (3H, s, 8-OCH_3_), 3.85 (3H, s, 4-OCH_3_), 3.81 (2H, S, H-1′), 2.88 (1H, m, 3′), 1.22 (3H, s, 3′-OCH_3_), 1.21 (3H, s, 3′-OCH_3_); ^13^C-NMR (150 MHz, CDCl_3_, *δ*, ppm), *δ*_C_ 211.9 (C-2′), 164.7 (C-2), 161.8 (C-4), 149.0 (C-8), 131.1 (C-8a), 122.7 (C-6), 119.9 (C-4a), 118.0 (C-3), 116.1 (C-5), 114.0 (C-7), 62.1 (4-OCH_3_), 56.8 (8-OCH_3_), 41.3 (C-3′), 37.2 (C-1′), 35.7 (1-NCH_3_), 18.5 (3′-CH_3_), 18.5 (3′-CH_3_).

### 3.4. Antitumor Bioassay

The in vitro cytotoxic activity of the compounds against MCF-7, HepG2, and PANC-1 was analyzed using the CCK-8 assay with Adriamycin as a positive control (Figure 5). The results demonstrated that none of the tested compounds exhibited significant antitumor activity compared to Adriamycin (Table 3). Although compound **3** demonstrated the best antitumor activity against MCF-7, with an IC_50_ value of 51.9 µM, it displayed weak activity against the other two tumor cells. The compounds tested showed considerable differences in their IC_50_ values compared to the positive control, indicating that the compounds do not have significant antitumor activity. The IC_50_ values for Adriamycin were found to be 2.03 µM, 3.85 µM, and 3.44 µM when tested on MCF-7, HepG2, and PANC-1, respectively. Additionally, the inhibitory effect of Adriamycin on tumor cells was observed to be dose-dependent, as the cell viability decreased with the increase in the drug concentration. For the assay, the concentrations of Adriamycin were set at 0.2, 0.5, 1, 2, 5, 10, 20, 50, and 100 µM, whereas for the tested compounds, they were set at 5, 20, 50, 100, and 200 µM for MCF-7 tumor cells and 50, 100, and 200 µM for HepG2 and PANC-1 tumor cells.

## 4. Discussion

Endophytic fungi are an important source for discovering naturally active substances [45]. In this study, a new endophytic fungus, viz., *Amphisphaeria orixae*, was isolated from the medicinal plant *O. japonica*. The fungi in *Amphisphaeria* are usually saprobes on woody branches and some monocotyledons, including grasses in terrestrial, mangrove, and freshwater habitats [21,22,23,24,30]. It is noteworthy that *A. orixae* is the first endophyte and first hyphomycetous asexual morph in *Amphisphaeria*, which broadens the morphological characteristics and lifestyles of this genus. The asexual sporulation of the new species *Amphisphaeria orixae* is relatively complex and includes phialidic and thallic conidiogenesis. This discovery is important for future taxonomic studies on this genus, as the generic boundaries of the family Amphisphaeriaceae have traditionally been based on sexual characteristics, and the delimitation of genera is still ambiguous and challenging. Furthermore, thirteen secondary metabolites, including a new isocoumarin, were isolated from the rice fermentation product of this fungus. All 13 compounds obtained in this study were isolated from the genus *Amphisphaeria* for the first time. Among them, compounds **1**–**4** are coumarins, and compounds **5**–**13** are quinoline alkaloids. The experimental strain was isolated from the roots of *O. japonica*. Interestingly, compounds **5**, **7**–**9**, and **12**–**13** were also obtained from the roots of the *O. japonica*, as reported by Liu et al. and Huang et al. [10,46], which provided evidence that the endophytic fungus is able to produce the same or similar compounds as the host. Moreover, these compounds are known to have significant antifungal, insecticidal, and food-repelling activities from the reported literature, with kokusaginine (**5**) having LC_50_ values of 16.66µg/mL and 5.32µg/mL against *Bursaphelenchus xylophilus* and *Meloidogynein congnita*, respectively, and lunidonine (**7**) not only killed *Meloidogynein congnita* and *Anopheles sinensis* but also showed a strong food-repelling effect on *Ostrinia furnacalis* [10,46]. These findings indicate that *Amphisphaeria orixae* may be a promising source of bioactive compounds with potential applications in agriculture and medicine. In summary, the discovery of this new endophytic fungus and its secondary metabolites further highlights the importance of exploring the microbial diversity of medicinal plants for the discovery of novel biologically active compounds.

## Figures and Tables

**Figure 1 microorganisms-11-01268-f001:**
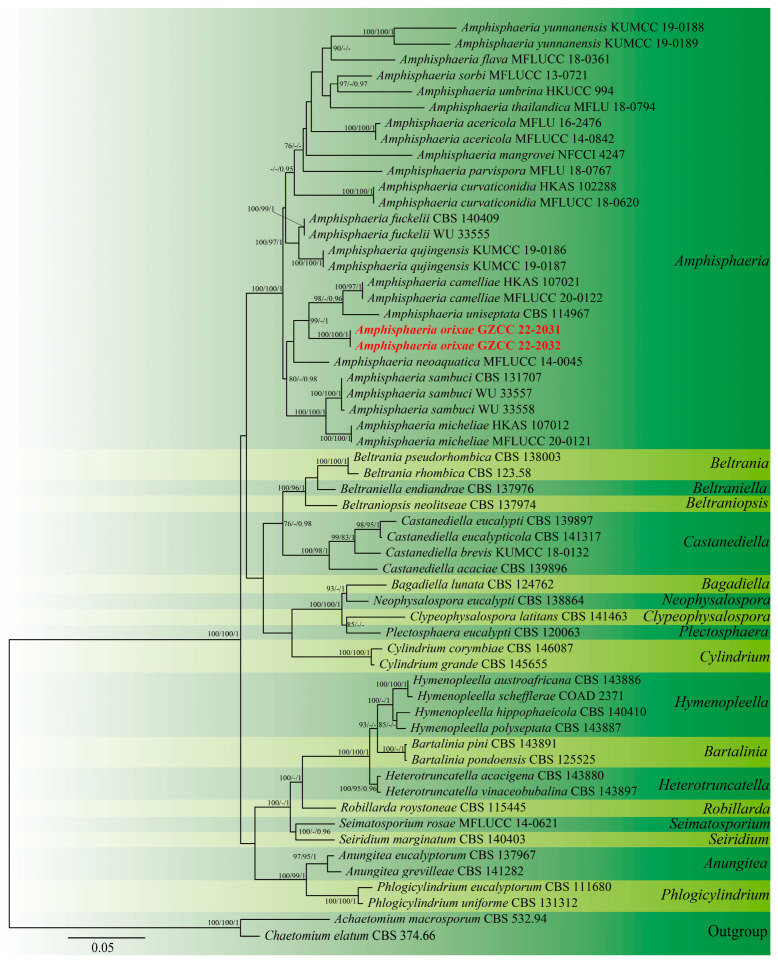
Maximum likelihood analysis of the phylogenetic tree constructed from the concatenated alignment of the ITS and LSU sequence data. Maximum likelihood (MLBS), maximum parsimony (MPBS), and Bayesian posterior probability (PP) bootstrap support values equal to or greater than 75% and 0.95, respectively, are provided close to the nodes. Bold and red are sequences that were recently generated.

**Figure 2 microorganisms-11-01268-f002:**
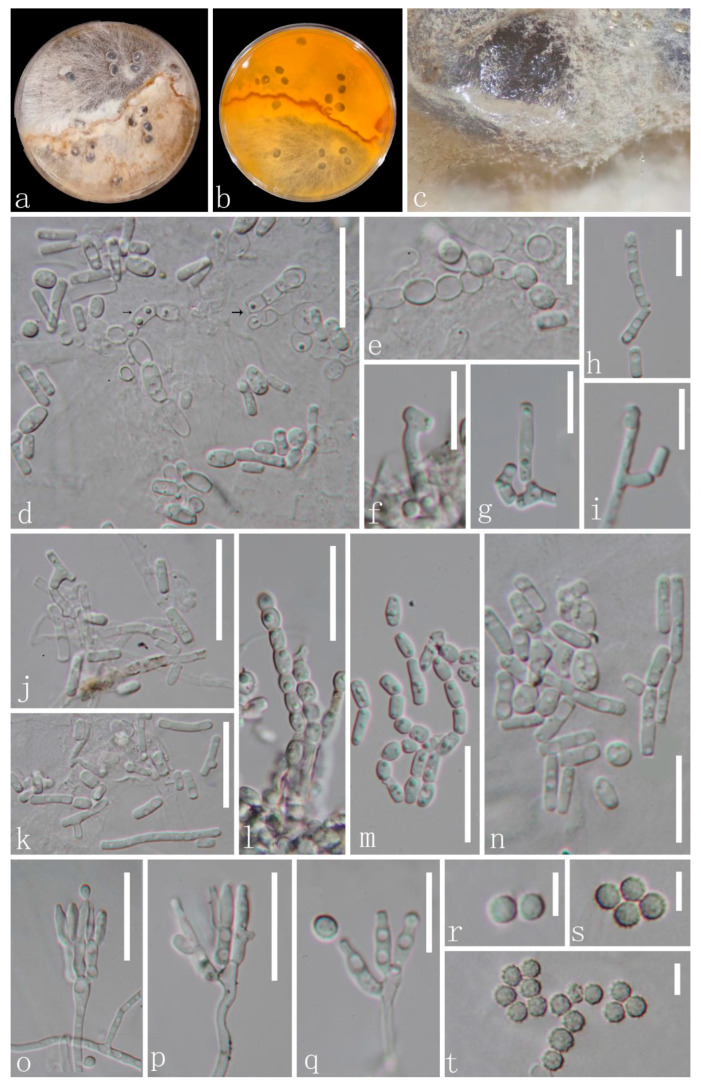
*Amphisphaeria orixae* (GZAAS 22-2031, Holotype). (**a**–**c**) Colonies on PDA mixed with polypropylene; (**d**–**n**) conidiophores, thallic conidiogenesis; (**e**,**l**) alternate-arthric; (**h**–**k**,**m**–**n**) thallic-arthric) and conidia; (**o**–**q**) conidiophores with conidiogenous cells (blastic) and conidia; (**r**–**t**) conidia. Scale bars: (**e**–**i**,**q**) = 10 μm, (**d**,**j**–**p**) = 20 μm; (**r**–**t**) = 5 μm.

**Figure 3 microorganisms-11-01268-f003:**
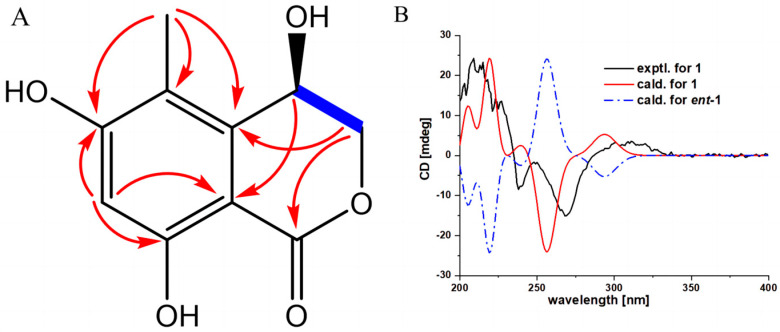
The key HMBC and H–H COSY correlations and experimental and calculated ECD curves for compound **1.** (**A**) HMBC correlations of compounds **1**; (**B**) Calculated and experimental ECD spectra of compound **1**.

**Figure 5 microorganisms-11-01268-f005:**
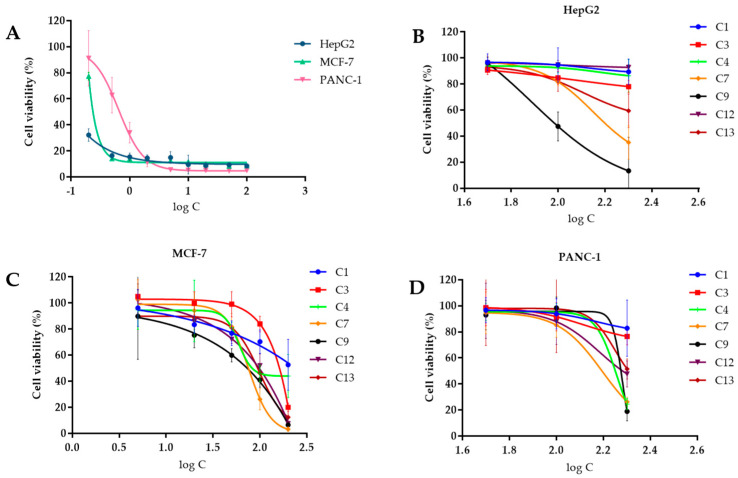
Effect of compounds on the cell viability of different tumor cells. (**A**) Effect of Adriamycin on the cell viability of three tumor cells; (**B**) effect of compounds on the cell viability of HePG2 tumor cells; (**C**) effect of compounds on the cell viability of MCF–7 tumor cells; (**D**) effect of compounds on the cell viability of PANC–1 tumor cells.

**Table 1 microorganisms-11-01268-t001:** GenBank accession numbers of the DNA sequences in this study and the taxa used.

Taxon	Strain	Status	GenBank Accessions	
			LSU	ITS
*Achaetomium macrosporum*	CBS 532.94		KX976699	KX976574
*Amphisphaeria acericola*	MFLU 16-2476	T	MK640424	MK640423
*Amphisphaeria acericola*	MFLUCC 14-0842	T	MF614131	MF614128
*Amphisphaeria camelliae*	HKAS 107021	T	MT756615	MT756621
*Amphisphaeria camelliae*	MFLUCC 20-0122		MT756616	MT756622
*Amphisphaeria curvaticonidia*	HKAS 102288		MT756618	MT756624
*Amphisphaeria curvaticonidia*	MFLUCC 18-0620	T	MT756617	MT756623
*Amphisphaeria flava*	MFLUCC 18-0361	T	MH971234	MH971224
*Amphisphaeria fuckelii*	WU 33555		KT949903	KT949903
*Amphisphaeria fuckelii*	CBS 140409	T	KT949902	KT949902
*Amphisphaeria mangrovei*	NFCCI 4247	T	MG844275	MG844283
*Amphisphaeria micheliae*	HKAS 107012	T	MT756619	MT756625
*Amphisphaeria micheliae*	MFLUCC 20-0121		MT756620	MT756626
*Amphisphaeria neoaquatica*	MFLUCC 14-0045	T	MK835805	MK828607
** *Amphisphaeria orixae* **	**GZCC 22-2031**	**T**	**OQ064541**	**OQ064543**
** *Amphisphaeria orixae* **	**GZCC 22-2032**		**OQ064542**	**OQ064544**
*Amphisphaeria parvispora*	MFLU 18-0767	T	MW240574	MW240644
*Amphisphaeria qujingensis*	KUMCC 19-0186		MN707567	MN707568
*Amphisphaeria qujingensis*	KUMCC 19-0187	T	MN55631	MN477033
*Amphisphaeria sambuci*	CBS 131707	T	KT949904	KT949904
*Amphisphaeria sambuci*	WU 33557		KT949905	KT949905
*Amphisphaeria sambuci*	WU 33558		KT949906	KT949906
*Amphisphaeria sorbi*	MFLUCC 13-0721	T	KP744475	KR092797
*Amphisphaeria thailandica*	MFLU 18-0794	T	MH971235	MH971225
*Amphisphaeria umbrina*	HKUCC 994		AF452029	AF009805
*Amphisphaeria uniseptata*	CBS 114967		MH554197	-
*Amphisphaeria yunnanensis*	KUMCC 19-0188	T	MN556306	MN477177
*Amphisphaeria yunnanensis*	KUMCC 19-0189		MN550992	MN550997
*Anungitea eucalyptorum*	CBS 137967	T	KJ869176	NR_132904
*Anungitea grevilleae*	CBS 141282	T	KX228304	NR_154719
*Bagadiella lunata*	CBS 124762	T	GQ303300	GQ303269
*Bartalinia pini*	CBS 143891	T	MH554330	MH554125
*Bartalinia pondoensis*	CBS 125525	T	MH875078	MH863602
*Beltrania pseudorhombica*	CBS 138003	T	NG_058667	MH554124
*Beltrania rhombica*	CBS 123.58	T	MH869260	MH857718
*Beltraniella endiandrae*	CBS 137976	T	MH878615	NR_148073
*Beltraniopsis neolitseae*	CBS 137974	T	MH878610	NR_148072
*Castanediella acaciae*	CBS 139896	T	MH878661	NR_137985
*Castanediella brevis*	KUMCC 18-0132	T	MH806358	MH806361
*Castanediella eucalypti*	CBS 139897	T	MH878665	KR476723
*Castanediella eucalypticola*	CBS 141317	T	MH878217	NR_145254
*Chaetomium elatum*	CBS 374.66		MH870466	KC109758
*Clypeophysalospora latitans*	CBS 141463	T	KX820261	KX820250
*Cylindrium corymbiae*	CBS 146087	T	MT223887	MT223792
*Cylindrium grande*	CBS 145655	T	MK876425	MK876384
*Heterotruncatella acacigena*	CBS 143880		MH554295	MH554084
*Heterotruncatella vinaceobubalina*	CBS 143897		MH554341	MH554139
*Hymenopleella austroafricana*	CBS 143886		MH554320	MH554115
*Hymenopleella hippophaeicola*	CBS 140410	T	MH878678	NR_154078
*Hymenopleella polyseptata*	CBS 143887		MH554321	MH554116
*Hymenopleella schefflerae*	COAD 2371	T	MH084761	MH128360
*Neophysalospora eucalypti*	CBS 138864	T	KP004490	KP004462
*Phlogicylindrium eucalyptorum*	CBS 111680		KF251707	KF251204
*Phlogicylindrium uniforme*	CBS 131312	T	JQ044445	JQ044426
*Plectosphaera eucalypti*	CBS 120063		DQ923538	DQ923538
*Robillarda roystoneae*	CBS 115445	T	MH874545	KR873254
*Seimatosporium rosae*	MFLUCC 14-0621	T	MH823070	LT853105
*Seiridium marginatum*	CBS 140403	T	MH878679	NR_156602

Note: Newly generated sequences are highlighted in bold, “T” indicates a type strain, and “-” indicates that were data not available in GenBank.

**Table 2 microorganisms-11-01268-t002:** ^1^H- and ^13^C-NMR spectral data of compound **1** (*δ* in ppm, *J* in Hz) in CD_3_OD.

Position	δ_H_ (*J* in Hz)	δc, Type
1		171.2 C=O
3	4.59 dd (12.6, 2.4) 4.48 dd (12.6, 2.4)	74.3 CH_2_
4	4.89 dd (2.4, 2.4)	62.1 CH
5		116.4 C
6		164.9 C
7	6.35 s	103.0 C
8		163.4 C
9	2.16 s	10.1 CH_3_
4a		140.3 C
8a		100.7 C

**Table 3 microorganisms-11-01268-t003:** Cytotoxicity of compounds on three tumor cell lines (half-maximal inhibitory concentration (IC50): µM).

Compound	HePG2	MCF-7	PANC-1
**1**	136.8	97.2	149.0
**3**	116.7	51.9	130.8
**4**	151.4	59.2	179.8
**7**	141.7	75.6	156.8
**9**	77.69	78.9	190.2
**12**	113.4	104.2	152.0
**13**	128.6	98.9	177.0
Adriamycin	3.85	2.03	3.44

## Data Availability

All data analyzed in this study are available within the manuscript and are available from the corresponding authors upon request.

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
