# Peer review of "Insights from the Endophytic Fungi in Amphisphaeria (Sordariomycetes): A. orixae sp. nov. from Orixa japonica and Its Secondary Metabolites"

_microorganisms, 2023, doi:10.3390/microorganisms11051268_

Round 1
Reviewer 1 Report
The manuscript presented a novel endophytic fungus from the roots of a medicinal plant (Orixa japonica). The authors introduced it as a new species (Amphisphaeria orixae) based on morphological evidences and molecular phylogenetic. A new compound and 12 known compounds were isolated from the culture of this fungus. Therefore, this work is important for microbiology and for the discovery of new natural products.
Correct "as sociations" in the abstract (2nd line).
The quality of English language is adequate.
Author Response
The manuscript presented a novel endophytic fungus from the roots of a medicinal plant (Orixa japonica). The authors introduced it as a new species (Amphisphaeria orixae) based on morphological evidences and molecular phylogenetic. A new compound and 12 known compounds were isolated from the culture of this fungus. Therefore, this work is important for microbiology and for the discovery of new natural products.
Point 1: Correct "as sociations" in the abstract (2nd line).
Response 1: We are very sorry that we described the wrong words in the manuscript, based on your comment, we have corrected the incorrect words in the manuscript (line 16). And we double-checked the rest of the article and corrected any grammatical errors and unreasonable content in the manuscript.Thanks.
Reviewer 2 Report
Review of the manuscript “Insights from the endophytic fungi in Amphisphaeria (Sordariomycetes): A. orixae sp. nov. from Orixa japonica and its secondary metabolites” submitted for consideration to the MDPI Microorganisms journal. In this paper authors studied fungal endophytes from roots of the medicinal plant, Orixa japonica, and reported several interesting findings. They found a new species in a genus of Amphisphaeria and supported their finding with morphological and phylogenetic analysis. They identified 13 compounds from a fermented substrate, including one unknown isocumarin. Some of these compounds may possess antifungal, insecticidal, and insect repelling properties. However, antitumor activities were not found. Overall, the manuscript is well written and needs just a few grammar touch ups. I have several suggestions and comments:
-
In the Abstract, say that antitumor activities were not observed.
-
“Each plant contains one or more endophytic microorganisms, which can produce the same or similar compounds as the host plant, and…” This sentence needs modification. It is not true that endophytes can produce only the same compounds as their host plants. Moreover, often only the endophyte but not its host-plant is responsible for insecticidal, insect deterring, or medicinal properties, see papers on Epichloë spp, endophytes from Echinacea purpurea, and more.
-
“In addition, as a natural plant, O. japonica can be combined with antibiotics…” This sentence needs improvement.
-
In the Methods, specify how many plants were sampled. Did you find A. orixae from each plant?
-
“The healthy materials were washed with running tap water at least 2 hours.” Specify that plant roots or leaves or something else was used for this.
-
“Sporulation was induced by adding polypropylene (PP) to potato dextrose agar (PDA) medium and variable temperature (alternating between 24 hours at 28 °C and 24 hours at 4 °C) with sweeping mycelium.” Needs improvement.
-
I didn’t find in the Methods how you extracted various compounds from the plant roots.
-
In the Results, could you mention if you have isolated fungi other than A. orixae.
-
“Culture characteristics: Colonies entire, superficial, flat, growing slowing on the PDA,..” Do you mean slow growing?
-
“Notes: Amphisphaeria orixae is the first hyphomycetous asexual morph reported in this genus based on molecular DNA data, which was morphologically differs from all existing.” Do you mean different?
-
Figures 3 and 4 may be combined in one figure.
-
It will be good to have some form of a summary of all compounds described. You can make a table or add their names to Fig. 5.
-
More clarification needed to the 3.4 Antitumor Bioassay section. Describe what your positive control tests with Adriamysin showed. In some cases with compounds 1-13 tests, cell viability was below IC50. Why did you consider this as a not-significant result? Why is 50uM concentration considered as a threshold?
-
Figures 6-9 may be organized in one figure. Move all the concentrations listed in a text to a figure legend.
-
For the Discussion section, I have a curiosity question: is it possible that in a host-plant A. orixae may have a different morphology and sexual reproduction than in a plate culture?
-
“Interestingly, compounds 5, 7-9 and 12-13 have also been obtained from the roots of the original plant, which provided the evidence for the endophytic fungus being able to produce the same or similar compounds as the host.” I doubt that these compounds were produced by the plant itself. Did you test roots from the plant without endophyte?
-
“Besides, these compounds showed significant antifungal, insecticidal and food-repelling activities.” Describe this in more detail.
A few sentences needs to be improved, see my comments above.
Author Response
Reviewer 2:
Review of the manuscript “Insights from the endophytic fungi in Amphisphaeria (Sordariomycetes): A. orixae sp. nov. from Orixa japonica and its secondary metabolites” submitted for consideration to the MDPI Microorganisms journal. In this paper authors studied fungal endophytes from roots of the medicinal plant, Orixa japonica, and reported several interesting findings. They found a new species in a genus of Amphisphaeria and supported their finding with morphological and phylogenetic analysis. They identified 13 compounds from a fermented substrate, including one unknown isocoumarin. Some of these compounds may possess antifungal, insecticidal, and insect repelling properties. However, antitumor activities were not found. Overall, the manuscript is well written and needs just a few grammar touch ups. I have several suggestions and comments:
Point 1: In the Abstract, say that antitumor activities were not observed.
Response 1: According to your comment, we expressed the results that the compounds had no significant antitumor activity in the abstract (line 29). Thanks.
Point 2: “Each plant contains one or more endophytic microorganisms, which can produce the same or similar compounds as the host plant, and…” This sentence needs modification. It is not true that endophytes can produce only the same compounds as their host plants. Moreover, often only the endophyte but not its host-plant is responsible for insecticidal, insect deterring, or medicinal properties, see papers on Epichloë spp, endophytes from Echinacea purpurea, and more.
Response 2: According to your comment, we revised the description of the sentence (line 45-49). In addition, we would like to thank you for knowledge us about endophytic fungi, for which we have reviewed more information, which is important for us to continue our research on endophytes in the future.
Point 3: “In addition, as a natural plant, O. japonica can be combined with antibiotics…” This sentence needs improvement.
Response 3: According to your comment, we revised the description of the sentence (line 71-73). Thanks.
Point 4: In the Methods, specify how many plants were sampled. Did you find A. orixae from each plant?
Response 4: We randomly collected multiple samples, but did not record detailed sample quantities. A. orixae was isolated from different roots of this plant and did not isolate this fungus from other parts of this plant.
Point 5: “The healthy materials were washed with running tap water at least 2 hours.” Specify that plant roots or leaves or something else was used for this.
Response 5: Thank you for your suggestions. We made a mistake, actually the fresh and healthy materials, including roots, stems and leaves, were washed with running tap water for at least 10 mins.
Point 6: “Sporulation was induced by adding polypropylene (PP) to potato dextrose agar (PDA) medium and variable temperature (alternating between 24 hours at 28 °C and 24 hours at 4 °C) with sweeping mycelium.” Needs improvement.
Response 6: According to your comment, we revised the description of the sentence (line 116-119). Thanks.
Point 7: I didn’t find in the Methods how you extracted various compounds from the plant roots.
Response 7: In our research, we extracted and isolated the compounds from the fermentation products of the fungius, rather than from the plant itself. We obtained compounds from fungal rice ferments and identified them, then, as a comparion, we searched for published literature, but not from the collected plants.
Point 8: In the Results, could you mention if you have isolated fungi other than A. orixae.
Response 8: Thank you for your suggestion. Regarding other fungi isolated from the same plant, our article mainly focuses on the species and secondary metabolites of the endophytic fungus Amphisphaeria orixae, which was identified in this study. However, we would like to mention that our previously published paper Lu et al. [6] isolated the novel endophytic fungus Diaporthe orixae and reported a new host record for Diaporthe caryae, both of which were also isolated from this plant.
Point 9: “Culture characteristics: Colonies entire, superficial, flat, growing slowing on the PDA ..” Do you mean slow growing?
Response 9: According to your comment, we correct this sentence as “Colonies entire, superficial, flat, slow growing on the PDA” (line 297). Thanks.
Point 10: “Notes: Amphisphaeria orixae is the first hyphomycetous asexual morph reported in this genus based on molecular DNA data, which was morphologically differs from all existing.” Do you mean different?
Response 10: Yes, the morphology of this species is distinct from all known species in this genus.
Point 11: Figures 3 and 4 may be combined in one figure.
Response 11: According to your comment, we have combined Figures 3 and 4 (line 341-343), and in the process of modifying them we also found an error in Table 2 and modified it. we appreciate this comment as we have previously debated whether to separate or combine them.
Point 12: It will be good to have some form of a summary of all compounds described. You can make a table or add their names to Fig. 5.
Response 12: According to your comment, we revised them (Figure 4). Thanks.
Point 13: More clarification needed to the 3.4 Antitumor Bioassay section. Describe what your positive control tests with Adriamysin showed. In some cases with compounds 1-13 tests, cell viability was below IC50. Why did you consider this as a not-significant result? Why is 50uM concentration considered as a threshold?
Response 13: Thank you for your suggestions. We rewrite this paragraph as “In vitro cytotoxic activity of the compounds against MCF-7, HepG2, and PANC-1 was analyzed using the CCK-8 assay with adriamycin as a positive control (Figure 6). The results demonstrated that none of the tested compounds exhibited significant antitumor activity compared to adriamycin (Table 3). Although compound 3 demonstrated the best antitumor activity with an IC50 value of 51.9 µM against MCF-7, it displayed weak activity against the other two tumor cells. The compounds tested showed a considerable difference in IC50 values compared to the positive control, indicating that the compounds do not have significant antitumor activity.”
Point 14: Figures 6-9 may be organized in one figure. Move all the concentrations listed in a text to a figure legend.
Response 14: According to this comment, we have combined Figures 6-9 in one figure and replaced the original bar chart with a line chart (line 459-463).
Point 15: For the Discussion section, I have a curiosity question: is it possible that in a host-plant A. orixae may have a different morphology and sexual reproduction than in a plate culture?
Response 15: We have attempted varous methods to promote spore production but we only observed the hyphomycetous morphology in this study. According to the published literature, endophytic fungi are capable of obtaining sexual morphs in culture medium, however, in this study, we were unable to observe any sexual morphs.
Point 16: “Interestingly, compounds 5, 7-9 and 12-13 have also been obtained from the roots of the original plant, which provided the evidence for the endophytic fungus being able to produce the same or similar compounds as the host.” I doubt that these compounds were produced by the plant itself. Did you test roots from the plant without endophyte?
Response 16: Due to the significantly lower quantities of endophytic fungi as compared to the plant tissue itself, the compounds isolated from plant tissue should originate from the plant itself. However, endophytic fungi may promote the production of specific compounds in plants. Nevertheless, it becomes challenging to isolate metabolites from plant tissue in the absence of endophytic fungi. We revised this sentence as “Interestingly, compounds 5, 7-9 and 12-13 have also been obtained from the roots of original plant based on Liu et al and Huang et al [10, 46], which provided the evidence for the endophytic fungus being able to produce the same or similar compounds as the host”. This result was obtained by comparing with the published studies.
Point 17: “Besides, these compounds showed significant antifungal, insecticidal and food-repelling activities.” Describe this in more detail.
Response 17: According to your comment, we describe this sentence in more detail at the Discussion (line 491-496). Thanks.